# Exposure to Gold Induces Autoantibodies against Nuclear Antigens in A.TL Mice

**DOI:** 10.3390/biology13100812

**Published:** 2024-10-11

**Authors:** Sara Puente-Marin, Said Havarinasab

**Affiliations:** 1Division of Inflammation and Infection (II), Department of Biomedical and Clinical Sciences (BKV), Linköping University, 581 83 Linköping, Sweden; sara.puentes@liu.se; 2Division of Clinical Chemistry and Pharmacology (KKF), Department of Biomedical and Clinical Sciences (BKV), Linköping University, 581 83 Linköping, Sweden

**Keywords:** autoimmunity, gold, A.TL mice, A.SW mice, ANA, ANoA, dsDNA, chromatin, histones, RNP

## Abstract

**Simple Summary:**

This study investigates the autoimmune effects of gold aurothiomalate (AuTM) exposure in genetically different mouse strains to understand how gold triggers autoimmune diseases. We focused on two strains, A.TL and A.SW, which share genetic similarities except for H-2-related genes. Here, we show that A.TL mice, when exposed to AuTM, developed specific immune responses, including increased levels of antinuclear antibodies (ANA) compared to A.SW mice. A.TL mice showed a particularly strong response, especially in females, resembling symptoms of human autoimmune diseases like lupus. These findings suggest that A.TL mice could be a valuable model for studying environment-induced autoimmunity, helping to explore the genetic mechanisms behind these conditions and potentially leading to improved diagnosis and treatment of autoimmune diseases in humans.

**Abstract:**

To demonstrate causation or/and assess pathogenic mechanisms of environment-induced autoimmunity, various animal models that mimic the characteristics of the human autoimmune diseases need to be developed. Experimental studies in mice reveal the genetic factors that contribute to autoimmune diseases. Here, the immune response of two mouse strains congenic for non-H-2 genes, A.TL (H-2^tl^) and A.SW (H-2^s^), was evaluated after 15 weeks’ exposure to gold aurothiomalate (AuTM). AuTM-treated A.TL mice showed anti-nuclear antibodies (ANA) with homogenous and/or fine speckled staining patterns and serum autoantibodies to ds-DNA, chromatin, histones, and ribonucleoproteins (RNP). Female A.TL mice showed a stronger immune response than males, as well as an increase of B cells in their spleen after 15 weeks of gold exposure. A.SW exposed for AuTM showed the induction of anti-nucleolar antibodies (ANoA) with a clumpy staining pattern, as well as an increase in splenic B and T cells. The serum autoantibodies levels in A.SW mice were limited compared to those of A.TL mice. Overall, A.TL presents a stronger immune response after gold exposure than A.SW. The immune response developed in A.TL presents similarities with the clinical manifestations in human autoimmune diseases. Thus, gold-exposed A.TL could constitute a potential experimental mouse model for the study of autoimmunity.

## 1. Introduction

Autoimmune diseases are a serious global and clinical problem that include a diverse range of disorders that can vary in their clinical symptoms and the organs involved [1,2,3]. Many of them are characterized by the development of antinuclear autoantibodies (ANA) [4].

Earlier, mutations in the immune system started the understanding of the immune effect. Modulations of the immune system in experimental mice, like neonatal thymectomy, single-gene deletion, or overexpression of gene(s), has been used to understand the effect of autoimmune disorders. Other approaches include the use of animal models that spontaneously develop autoimmune diseases, or the study of the immune response in animals and humans exposed to xenobiotics such as drugs, pristane, or metals [5].

Human drugs, such as procainamide and hydralazine, act through DNA hypomethylation or the formation of reactive metabolites that trigger ANA [6]. In mice, hydrocarbon oils, particularly pristane, have been found to trigger the development of ANA against many cell components mimicking the diversity of autoantibodies in systemic lupus erythematosus (SLE) [7]. Xenobiotics, like metals, may also induce ANA and systemic autoimmunity in humans [8] and animal models [9].

In mouse models, exposure to mercury [10], silver [11,12,13], and gold [14,15] in different forms may develop ANA in susceptible genotypes, and more specifically, anti-nucleolar antibodies (ANoA). ANA/ANoA have also been observed in a subset of patients with autoimmune diseases like systemic scleroderma (SSc), SLE, and Raynaud phenomenon [16,17], and in mercury-exposed artisanal gold miners [8]. Silver and gold cause a modest activation of the immune system with limited tissue immune complex (IC) deposition in mice [12,14], while mercury causes a more extensive activation of the autoimmune system with lymphadenopathy, hypergammaglobulinemia [10], and systemic IC deposits [18]. The susceptibility to metal-induced autoimmunity in mice is controlled genetically by both the H-2 (mouse major histocompatibility complex) genes [12,14,19], as well as by genes outside the H-2 region [20].

Global heavy metal pollution may trigger or exacerbate autoimmune diseases in humans [21]. The mining industry (gold and mercury) [8,22], fish consumption (mercury) [23,24], wound care products, medical devices, and silver (Ag) nanoparticles (AgNPs) (silver) [25,26] lead to heavy metals exposure for humans. Several metals interact with the immune system; one of them is gold. Gold in its metallic state Au(0) is inert, while the Au2+ and Au3+ forms are dissolved and react with proteins [27]. The main route of exposure to gold compounds in humans was in the treatment of rheumatic diseases. The reference treatment during the 20th century for rheumatoid diseases, especially for rheumatoid arthritis, was gold salts with 70% of patients responding to the anti-inflammatory effects of gold treatment [28]. However, these drugs could also cause adverse effects like thrombocytopenia, granulocytopenia, proteinuria, or even nephrotic syndrome in some patients [29,30,31]. It is known that patients with certain HLA-DR alleles are more prone to developing autoimmunity [32] or experiencing adverse effects after gold exposure [33]. These adverse effects could be found in one-third of patients, and eventually the treatment needed to be stopped [31]. Gold treatment was phased out at the turn of the century [28], and several new drugs, like monoclonal antibodies and biologics, have practically eliminated gold as an anti-rheumatic substance in the Western world [34]. Nevertheless, these clinical observations encouraged studies with gold in susceptible mouse models to better understand human autoimmune responses. Gold exposure in certain inbred mouse strains carrying the H-2^s^ complex, like A.SW, induces autoantibodies to fibrillarin, a nuclear protein target of autoantibodies commonly found in some patients with systemic autoimmune diseases [16,17]. However, the congenic strain A.TL, which carries the H-2^tl^ complex, does not develop anti-fibrillarin antibodies in response to gold, mercury, or silver; then, A.TL was called a resistant or intermediate ANA-positive strain. Therefore, no in-depth studies have been done beyond the presence or absence of ANoA. In this study, we present a more complete immune response evaluation of the ANoA-resistant strain A.TL after gold exposure, showing the induction of ANA IgG titers, anti-dsDNA, -RNP, -histones, and -chromatin antibodies, and comparing it with that of the ANoA-susceptible A.SW strain.

## 2. Material and Methods

### 2.1. Animals

Thirty A.TL (H-2^t1^) mice (15 male; 15 female) aged 12–16 weeks at the start of the study (originally obtained from Harlan Ltd., Bicester, Oxon, UK), and 20 A.SW (H-2^s^) mice (10 male; 10 female) aged 8–12 weeks at the start of the study, obtained from Taconic M&B (Ry, Denmark), were used. The configuration of the H-2 and the genetic background is shown in Table 1. All animals were housed and bred in a high-barrier unit in steel-wire cages under 12-h light-dark cycles and given sterilized food pellets and water ad libitum.

### 2.2. Gold (Aurothiomalate) Treatment

A.TL and A.SW mice were randomized into two groups: control and treated (Table 1).

Gold-treated A.TL and A.SW mice were injected intramuscularly, every third day for 15 weeks with 0.1 mL sodium aurothiomalate C_4_H_4_AuNaO_4_S (AuTM) (Myocrisin; Aventis Pharma AB, Stockholm, Sweden) containing 0.45 mg AuTM (22.5 mg/kg·bw) [15]. Control A.TL and A.SW mice were given tap water only and were maintained in the same condition as the treated mice.

### 2.3. Blood and Tissue Sampling

Blood samples were collected from all mice, through the retro-orbital plexus, before the start of the treatment (week 0), and after 5, 10, and 15 weeks. The blood samples were centrifuged at 500× *g* for 15 min, and the sera were separated. Upon sacrifice at week 15, tissue specimens from the left kidney and spleen were collected for examination of glomerular immune complex (IC) deposition and splenocytes surface markers. Serum and tissue were stored at −80 °C until analysis.

### 2.4. Analysis of Antinuclear Antibodies (ANA) by Indirect Immunofluorescence

The presence, pattern, and titer of antinuclear antibodies (ANA) in serum of A.TL and A.SW mice were detected by indirect immunofluorescence staining using HEp-2 (Human Epithelial) cells as substrate [14]. Briefly, sera were serially diluted 1:80–1:5120 in phosphate-buffered saline (PBS) and incubated on slides with a monolayer of HEp-2 cells (Binding Site Ltd., Birmingham, UK), followed by Fluorescein isothiocyanate (FITC)-conjugated goat anti-mouse IgG_total_ and IgG isotypes (IgG_1_, IgG_2a_, IgG_2b_, and IgG_3_) (1:50 dilution) (Southern Biotechnology Inc., Birmingham, AL, USA). The titer of ANA was defined as the highest serum dilution, which gave a nuclear staining. The absence of fluorescence staining by a serum diluted 1:80 was considered as a negative result. Samples that resulted negative for ANA IgG_total_ titer were not tested for IgG subclasses (no data, ND). The staining was assessed using a Nikon incident-light fluorescence microscope (Nikon Eclipse E600, Instech, Kanagawa, Japan). All examinations were done with coded samples. A pool of sera from young ANA-negative mice was used as the negative control. Serum from MRL-lpr/lpr mice was used as the positive control.

### 2.5. Serum Anti-dsDNA Antibodies Assessed by Indirect Immunofluorescence

Anti-double-stranded DNA (dsDNA) antibodies in mouse serum were detected using the *Crithidia luciliae* assay [36]. Slides with *Crithidia luciliae* (Binding Site) were incubated with mice sera diluted 1:10 in PBS. Bound anti-dsDNA antibodies were detected by FITC-conjugated goat anti-mouse IgG_total_ antibodies (Sigma, St. Louis, MO, USA) using coded samples. The staining was assessed using a Nikon incident-light fluorescence microscope (Nikon Eclipse E600, Instech, Kanagawa, Japan). A staining of the kinetoplast was recorded as positive dsDNA staining. A pool of sera from young ANA-negative mice was used as the negative control. Serum from NZB/NZW F1 mice was used as the positive control.

### 2.6. Serum Anti-Chromatin Antibodies Assessed by Enzyme-Linked Immunosorbent Assay (ELISA)

Serum anti-chromatin antibodies were measured as described previously [37]. Calf thymus chromatin was added to microtiter plates (Nunc, Copenhagen, Denmark). After overnight incubation at 4 °C, the plates were blocked with 0.1% gelatin overnight at 4 °C. Next, serum samples diluted 1:400 in PBS were added for 90 min at room temperature in duplicate. Using ALP-conjugated goat anti-mouse IgG antibodies (Caltag Laboratories, Burlingame, CA, USA) and pNPP (p-Nitrophenyl Phosphate) substrate, the optical density (OD) was measured at 405 nm and the background values were subtracted. A pool of sera from strong ANA-positive NZB/W F1 mice (homogeneous pattern), and a pool serum from weakly ANA-positive NZB/W F1 sera, were used as the positive controls. A pooled serum from young ANA-negative mice was used as the negative control.

### 2.7. Serum Anti-RNP Antibodies Assessed by Enzyme-Linked Immunosorbent Assay (ELISA)

Serum anti-RNP antibodies were assessed using an ELISA kit (Euroimmun, Lübeck, Germany). Microplates were coated with native proteins from U1 snRNP particle. After overnight incubation at 4 °C, serum samples diluted 1:200 in PBS were added for 30 min at room temperature in duplicate. After incubation, HRP-conjugated goat anti-mouse IgG antibody (Southern Biotechnology) was added. Following washes, the substrate was added, and the reaction stopped with 0·5 M citric acid. The OD was measured at 450 nm, and the background values were subtracted. Pooled serum from ANA-positive MRL-lpr/lpr mice and pooled serum from young ANA-negative mice were used as the positive and negative controls, respectively.

### 2.8. Serum Anti-Histone Antibodies Assessed by Enzyme-Linked Immunosorbent Assay (ELISA)

Serum anti-histone antibodies were determined using the Quanta Lite Histone ELISA kit (INOVA Diagnostics, Inc., St. Ingbert, Germany). Sera diluted 1:80 were added to the wells in duplicate and incubated for 30 min. After repeated washing steps, goat F(ab’)2 anti-mouse IgG (diluted 1:2000) was then added to the wells followed by a 30-min incubation. After washing the wells, TMB chromogen was added. After 30 min, the HRP stop solution was added. The optical density was measured at 450 nm, and the background values were subtracted. Pooled serum from ANA-positive MRL-lpr/lpr mice and pooled serum from young ANA-negative mice were used as the positive and negative controls, respectively.

### 2.9. Analysis of Cell Surface Markers of Splenocytes by Flow Cytometry

Single-cell suspensions from spleen tissue harvested from female A.TL and A.SW mice at week 15 was prepared as described previously [14]. The total splenic cell density was adjusted to 20 × 10^6^ cells/mL, and 50 µL (1 × 10^6^ cells) was incubated together with 40% rabbit serum (Dako, Copenhagen, Denmark) as an Fc-receptor blocking agent for 20 min at 4 °C. After washing the cells, fluorochrome-conjugated monoclonal antibodies (mAbs) were added to the cells as previously described. The following mAbs (Becton Dickinson (BD); San Diego, CA, USA) were used: conjugated with FITC-CD3 (pan-T cells, clone 145-2C11); peridinin-chlorophyll proteins (PerCP)-CD4 (T helper cell, clone RM4-5); APC-CD25 (activated T and regulatory T cells, clone PC61); APC-CD19 (pan-B cells, clone 1D3); PE-CD49b (pan-NK cells, clone DX5); and PE-CD90.2 (Thy-1.2) (peripheral T cells, clone 30-H12). To analyze cytoplasmic IFNγ^+^ cells (cIFN-γ^+^), cells were incubated with 250 μL of Cytofix/Cytoperm (BD) for 20 min at 4 °C, followed by two washes with HBSS-2% FCS buffer and then resuspension in 1 mL Perm/Wash (BD). The Alexa Fluor^®^647 anti-mouse IFN-γ (clone ZMG1.2) were diluted 1:50 in Perm/Wash solution and incubated with the cell suspensions for 30 min. Washing with HBSS-2% FCS buffer was repeated twice, and the pellet was resolved in 1 mL Perm/Wash. RPMI-1640, Fetal Calf Serum (FCS), and Hanks’ balanced salt solution (HBSS) were obtained from Gibco (Paisely, UK). Acquisition of 1 × 10^5^ cells were saved in list mode of Galios flow cytometer (Beckman Coulter, Brea, CA, USA). Data analysis was performed using the Kaluza software (version 2.1) (Beckman Coulter).

### 2.10. Glomerular Immunocomplex Deposits

Pieces of the left kidney were examined by immunofluorescence, as described before [19], using FITC-conjugated goat anti-mouse IgG (Southern Biotechnology) antibody. The titer of IgG_total_ was determined by serial dilution of the antibodies from 1:40 to 1:5120. The end-point titer of the deposits was defined as the highest dilution of antibody at which a fluorescence could be detected. When no fluorescence was detected at a dilution of 1:40, the result was recorded as ‘0’.

### 2.11. Statistics and Software

The differences between the controls and the AuTM-treated mice, and the differences between both strains, were analyzed by the Mann–Whitney U test. Fisher’s exact test was used to analyze serum ANA and serum anti-dsDNA antibodies positivity. A *p*-value less than 0.05 was considered statistically significant. *, **, ***, **** = *p* value < 0.05, 0.01, 0.001, 0.0001 was used for figures, and a, b, c, d = *p* value ≤ 0.05, 0.01, 0.001, 0.0001 was used for tables. Graphpad Prism 8.0.1 (GraphPad Software, La Jolla, CA, USA) was used for statistics and graphic representation. Clustering heatmap was constructed using ClustVis software (https://biit.cs.ut.ee/clustvis/, accessed on 3 October 2024) [38].

## 3. Results

### 3.1. Serum Antinuclear Antibodies (ANA/ANoA) Pattern, Titer, and IC Deposits

Treatment with AuTM in the two mouse strains with different H-2 haplotypes, *t1* (A.TL) and *s* (A.SW), resulted in different ANA specificity. The A.SW mice developed anti-nucleolar antibodies (ANoA), staining the nucleoli with a clumpy pattern (Figure 1A). The A.TL mice showed anti-nuclear antibodies (ANA), staining the nucleus with a homogeneous (Figure 1B) or fine speckled pattern (Figure 1C) [39].

In A.TL mice, AuTM treatment significantly increased the ANA titer of IgG_total_ at week 5, when 94% of mice (15 out of 16) showed ANA with homogenous (9) and fine speckled (6) pattern (Table 2). The titer of IgG_total_ continued increasing over time, and at weeks 10 and 15, 100% of A.TL mice (16) showed ANA in homogenous (13,12) and fine speckled (3,4) pattern, respectively (Table 2). At the end of the study, four out of 14 A.TL controls showed ANA at the lowest titer evaluated (1:80) in a fine speckled staining pattern. No ANA was detected before the onset of the treatment (week 0) in controls or AuTM-treated A.TL mice (Table 2).

In A.SW mice, AuTM treatment significantly increased the ANoA titer of IgG_total_ already at week 5 and continued increasing it until the end of the study (week 15). At week 5, 60% of AuTM-treated A.SW mice showed ANoA. This increased to 100% of mice in weeks 10 and 15. No ANoA was detected in the controls or before the onset of the treatment (week 0) in AuTM-treated A.SW mice (Table 2).

The IgG ANA titer increased with AuTM treatment time and compared with control mice in both strains (Table 2, Appendix A). However, this increase was more pronounced in A.TL compared to A.SW mice after AuTM treatment (Figure 2). IgG_total_ titers were significantly higher in A.TL already at week 5 and continued higher until week 15, compared to A.SW mice (Figure 2A). Likewise, higher levels of IgG ANA of the four subclasses were detected in AuTM-treated A.TL mice compared to A.SW mice, statistically significant for IgG_2a_ at weeks 10 and 15, IgG_2b_ at week 15 and IgG_3_ at week 10 (Figure 2B).

After 15 weeks of treatment with AuTM, A.TL mice significantly increase the titer of IgG_total_ deposition in the glomerulus compared to control mice. In A.SW mice, the titer of IgG_total_ increased slightly but not in a significant manner (Table 2).

### 3.2. Serum Autoantibodies against dsDNA, Chromatin, Histones, and RNP

Serum autoantibodies detected after treatment with AuTM presented differences when comparing both strains, A.SW and A.TL

After treatment with AuTM, only one out of 10 A.SW mice resulted positive for dsDNA at week 15. On the other hand, after AuTM exposure, 70% (11 out of 16) of A.TL mice were positive for dsDNA antibodies at weeks 10 and 15 (Table 2) and were statistically significant compared to AuTM-treated A.SW (Figure 3A). None of the AuTM-treated A.TL or A.SW mice were positive for dsDNA antibodies at the onset of the study, nor were any control mice during the study (Table 2). Likewise, the ELISA immunoassay showed that A.TL mice presented statistically higher levels of anti-chromatin, -histones, and -RNP antibodies than A.SW mice at all the time points analyzed during the whole study after AuTM treatment (Figure 3B).

In A.TL mice, serum anti-chromatin antibodies were statistically increased after 5 weeks of AuTM treatment compared to A.TL controls, and the levels continued increasing along the study (Figure 3B). Serum levels of anti-RNP and anti-histones in AuTM-treated A.TL were significantly higher than in controls at weeks 10 and 15 of the study (Figure 3B).

On the other hand, all AuTM-treated A.SW mice showed very low levels of anti-RNP and anti-histones during the whole study (Figure 3B). Only significantly higher levels of anti-chromatin antibodies were detected for A.SW mice after 10 and 15 weeks of AuTM treatment when compared to the control.

### 3.3. T- and B Cell Markers in Spleen Following AuTM Treatment

To assess the cellular effects of AuTM on T- and B lymphocytes, the total number of T and B cells, as well as cell surface markers for regulatory T cells (CD4^+^ CD25^+^), NK cells (CD3^+^ CD49b^+^ IFN-γ), and Thy-1.2, alloantigen (CD3^+^ CD90.2^+^), were analyzed in female A.TL and A.SW mice treated with AuTM and controls.

After 15 weeks of AuTM treatment, A.SW mice showed a significant increase in the number of splenocytes, in the total number of splenic T cells (CD3^+^), and in the number of B cells (CD19^+^) compared to A.SW controls. Significantly higher production of IFN-γ in splenocytes and higher Thy-1.2 alloantigen was detected in AuTM-treated A.SW mice compared to controls. However, no statistical differences were found in T-reg cells (CD4^+^, CD25^+^) in A.SW mice, but interestingly, the treatment with AuTM decreased slightly the number of the T-reg cells compared to controls (Table 3).

In the case of A.TL mice, a significant increase in the number of B cells (CD19^+^) was observed compared to controls, while the number of splenocytes and T cells (CD3^+^) increased slightly but not significantly. AuTM treatment did not affect the number of the T-reg, NK, and Thy-1.2, alloantigen cells in AuTM-treated A.TL mice compared to controls (Table 3).

To summarize, after AuTM treatment, A.SW mice showed an increase of T and B cell markers while A.TL mice showed an increase in B cell markers in splenocytes.

### 3.4. Differences in the Immune Response of A.TL Mice by Gender

The immune response in A.TL mice after AuTM treatment presented differences according to gender. The number of A.TL mice that develop ANA of IgG_total_ after AuTM treatment was similar in both females and males (Table 4). However, the ANA titer was significantly higher in females after 10 and 15 weeks of treatment (Figure 4A). Likewise, females showed higher titer of IgG subclasses than males. Statistically significant were ANA titers of IgG_1_ at week 5, IgG_2a_ at weeks 10 and 15, IgG_2b_ at week 10, and IgG_3_ at week 15 (Figure 4B).

By analyzing gender differences in serum antibody levels after AuTM treatment using ELISA, A.TL females showed higher levels of anti-chromatin during the study, statistically significant at weeks 0 and 5 compared to males (Figure 5A). Even though no statistical differences were found for serum anti-RNP and anti-histones antibodies by gender, males showed higher levels of anti-RNP than females along the study. A.TL males also showed higher levels of anti-histones antibodies until week 10 and then decreased at week 15, when levels were higher for A.TL females (Figure 5A). No statistical differences were found in the number of positive individuals for anti-dsDNA antibodies between A.TL males and females (Table 4). Comparing the glomerular immune deposits between A.TL females and males treated with AuTM at week 15, higher titers of IgG_total_ were observed in females, but they were not statistically significant (Figure 5B).

To summarize the differences between A.TL males and females after AuTM treatment, a multivariate analysis of serum antibodies and glomerular immune deposits, represented as a clustering heatmap, illustrates a different profile in A.TL females and males at the end of the study (week 15). The heatmap showed higher levels of antibodies for A.TL females at week 15, except for anti-RNP antibodies whose levels was higher in males (Figure 5C).

## 4. Discussion

The inbred mouse strain A.TL was reported to be resistant to mercury and gold because, even when restricted immune stimulation was found in previous studies [14,19,40,41], the lack of ANoA induction has limited further investigation. In this study, we report that gold induces immunological alterations in A.TL mice characterized by the development of ANA and increased levels of autoantibodies against dsDNA, chromatin, histones, and RNPs, compared with control A.TL and with A.SW strain. A.TL (H-2^tl^) and A.SW (H-2^s^) strains share the non-H-2 genes (the “A” background). After gold exposure, they developed different antibody specificities, while A.SW developed ANoA and A.TL induced ANA, as we report in this study. Then, the different specificity and immune responses found in the present study between both strains might be mapped to the H-2 region. The H-2A locus (Table 1) was reported to be a regulator of ANoA-susceptibility and resistance in A.SW and A.TL, respectively [19,42], while expression or non-expression of H-2E locus was associated with lower/higher ANoA, respectively [19]. The absence of surface expression of H-2E in A.SW mice exposed to gold had contributed to the development of ANoA in this strain [14].

A.SW mice is a well-known strain in murine metal-induced autoimmunity for the induction of ANoA against the nucleolar protein, fibrillarin, by mercury [19,41,43], silver, [12] and gold [14]. The mechanism of induction of anti-fibrillarin antibodies by gold has not been fully established but is thought to be analogous to the findings for mercury [44] and silver [45]. However, in A.SW mice exposed to gold, the induction of anti-fibrillarin antibodies requires longer times, and the titers are lower than in mercury [41] or silver [12]. The time needed for metabolizing gold ions might partly explain the delay in antibody production, but still other features of the metal-induced autoimmune syndrome in H-2^S^ mice differ between the three metals [14]. Most likely, mercury stimulates T cells to produce interleukins that may stimulate B cells in susceptible strains like A.SW, shifting towards Th2-type immune responses [41]. However, some studies showed that B cell activation cannot be fully attributed to this Th2 response, and further research is needed to fully elucidate the autoantibody response. Here, we found that, after gold exposure, the number of both T and B cells are increased in A.SW, although not comparable with the lymphoproliferation observed with mercury and silver [12,14,41]. Gold also induces limited and relatively low renal deposits of immune complexes compared to those of mercury [14]. Gold produces a moderate response compared to mercury in A.SW. However, the mechanisms through which gold stimulates the immune system need to be addressed. Despite the differences in metal-induced autoimmune syndrome, previously our group found that gold was still a potent interactor with the immune system of A.SW mice, increasing the number of splenocytes and developing ANoA [14]. Here, we also showed an increase of ANA titers after gold exposure in A.SW, in fact, higher that the titer reported previously by our group [14]. However, when comparing these ANA titers with those of A.TL mice, A.SW titers were significantly lower. As previously reported, only marginal titers of IgG were detected in renal glomerular mesangium of A.SW after gold treatment [14]. In addition, we went further in the analysis of autoimmune response of gold-treated A.SW mice, and we showed that the serum levels of anti-chromatin, -RNP, and -histones antibodies as well as the presence of antibodies against dsDNA were very low or not detectable.

Regarding A.TL mice, previous studies reported no ANoA, no lymphoproliferation, and only limited IC deposits after gold exposure [14,40]. After mercury exposure, a slight B cell response was found in A.TL mice [41]. Hultman et al. reported that exposure to mercury caused homogeneous ANA only in 63% of A.TL mice [19]. They stated that (a) A.TL is a low-responder strain for mercury-induced autoimmunity and (b) that the non-H-2 background genes influence the susceptibility in development of ANA, being the A strain background was especially susceptible [19,40]. In the present study, the number of B cells (CD19+) in gold-treated A.TL mice showed a significant increase. This contrasted with our previous observation using B220 as a B cell marker [14]. The difference may be explained by the fact that CD19 is a more selective marker for the B cell lineage than B220 [46]. However, no increase in the number of T cells or IFN-γ- producing cells was found, which might have explained the stimulation of B cells in A.TL as it is thought to occur in A.SW mice [41]. Thus, B cells stimulation in A.TL appears to be due to factors other than those in A.SW and still needs to be clarified. The increased titer of IgG found in renal immune deposits in A.TL after gold exposure is consistent with previously published data [14]. However, the titers are limited (approximately 10 times lower) compared to those of mercury [19]. The significant increase in the production of anti- RNP, -dsDNA, -chromatin, and -histones antibodies in A.TL exposed to gold, showed here, points out the interest of the A.TL strain for the study of human autoimmune diseases in which these antibodies have a crucial role in serological and clinical manifestations [47,48]. In addition, fine speckled and homogenous patterns found in A.TL mice after gold treatment are associated with autoimmune disorders in humans, such as SLE, juvenile idiopathic arthritis, chronic autoimmune hepatitis, and systemic autoimmune rheumatic diseases [39,49,50].

However, inbred mouse strains do not represent the genetic heterogeneity in humans. We recently reported that in outbred Swiss Webster (SW) mice, with a heterogeneous H-2 region, gold treatment led to ANA with a homogenous and speckled pattern, but not ANoA. SW mice also developed autoantibodies against histones, dsDNA, and chromatin in response to gold [51]. The shared features of the immune response to gold between SW and A.TL (mice with different H-2), and the differences between A.SW and A.TL (mice with different H-2), might indicate further evidence for the role of H-2 but also non-H-2 genes in the autoimmune response to heavy metals. It remains to elucidate profoundly the genetical differences within and outside the H-2 genes in xenobiotic-induced autoimmunity.

Since the antigen specificity of autoantibodies in gold-treated mice differs between congenic mouse strains, a comparison could also be made with the specificities in mouse strains that spontaneously develop autoimmunity. The A.TL immune response to gold also presents similarities with the lupus-prone strains BXSB, MRL/l, and NZB/NZW F1 that spontaneously develop elevated serum IgG levels, ANA titers, and anti-dsDNA, -RNP, -chromatin, and -histones antibodies [52,53,54,55,56,57,58,59]. We found that the development of ANA occurs earlier in gold-treated A.TL than in the lupus-prone strains [52,58]. Autoantibodies against chromatin and its components, histones, and dsDNA are characteristic of the human autoimmune disease SLE and drug-induced lupus, but the mechanisms of their induction remain unknown [59,60]. Another important characteristics of MRL/l and NZB/NZW F1 strains is the more accelerated disease process in females compared to males, that correlates with higher autoantibodies titers (ANA and anti-dsDNA predominantly) and higher rates of mortality [52,54,58,61]. Similarly to humans, lupus develops primarily in females, with more severity and percentage than in males. Previous studies have demonstrated that expression of autoimmunity in mice is modified by sex hormones [62,63,64,65]. We found here that A.TL females have a stronger immune response compared to males after 15 weeks of gold exposure. A.TL females present higher ANA titers of IgG_total_, IgG_2a,_ and IgG_3_ subclasses. Although not significant, females express higher levels of anti-chromatin, -dsDNA, and -histones antibodies and more glomerular immune-complex than A.TL males after 15 weeks of gold exposure.

## 5. Conclusions

Taken together, our data show an autoimmune response toward chromatin components and an increase of IgG-producing B cells in A.TL mice following gold exposure, which is further exacerbated in females. This autoimmune reaction is discernible as early as 5 weeks after exposure to gold. A.TL mice showed antibodies targeting nuclear antigens and higher titers of ANA in comparison to A.SW, another inbred strain with “A” genetic background due to different genetic regulation. In humans, serum anti-nuclear antibodies such as anti-dsDNA, -RNP, and -histones antibodies are specific markers for clinical diagnosis of SLE and other rheumatic diseases more commonly affecting females [66,67,68,69]. The role of these antibodies in pathogenesis of autoimmune disease still needs to be investigated to understand the mechanisms of disease and immune system dysfunction, notably using animal models, to ultimately discover new therapeutic targets. The development of anti-nuclear antibodies in A.TL mice after 5 weeks of gold exposure advocates further studies on A.TL as another candidate model to study autoimmunity.

## Figures and Tables

**Figure 1 biology-13-00812-f001:**
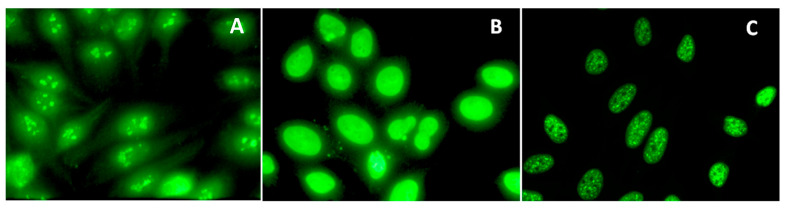
Serum anti-nuclear antibody pattern in AuTM-treated mice assessed by indirect immunofluorescence. Serum was incubated on HEp-2 cells and then detected with FITC-conjugated anti-mouse IgG antibodies. Following 15 weeks of AuTM treatment, 100% of A.SW mice developed ANoA with a clumpy staining pattern (**A**), while in A.TL mice, 75% developed homogeneous (**B**) and 25% fine speckled nuclear pattern (**C**).

**Figure 2 biology-13-00812-f002:**
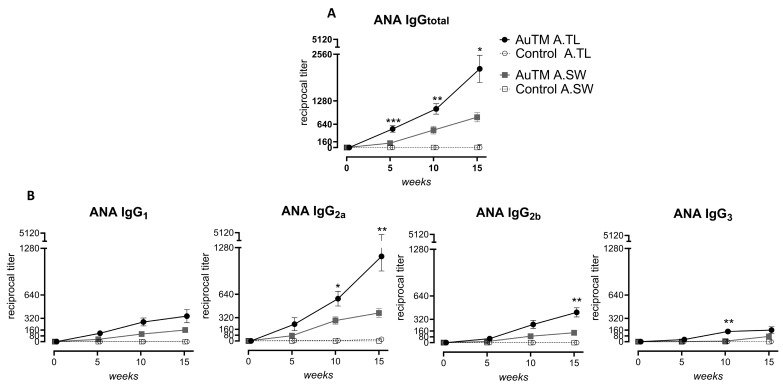
Titer of serum anti-nuclear antibodies (ANA). (**A**) IgG_total_ titer and (**B**) IgG subclass titers in A.TL and A.SW mice during 15 weeks of AuTM or H_2_O (control) treatment detected by indirect immunofluorescence staining of HEp-2 cells. Circles represent mean values of the A.TL strain treated with AuTM (black, n = 16) or control (empty circle, n = 14). Squares represent mean values of the A.SW strain treated with AuTM (gray, n = 10) or control (empty square, n = 9). Error bars indicate SEM. * = *p* < 0.05; ** = *p* < 0.01; *** = *p* < 0.001 indicate differences between A.TL and A.SW mice treated with AuTM using the Mann–Whitney U test.

**Figure 3 biology-13-00812-f003:**
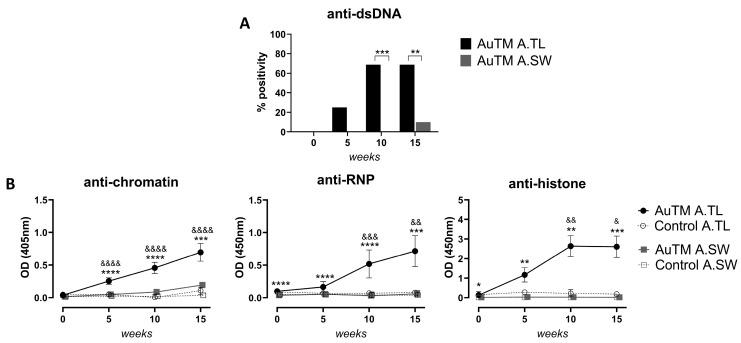
Serum autoantibodies against nuclear antigens. (**A**) Percentage of mice positive for anti-dsDNA antibodies during 15 weeks of AuTM treatment detected by *Crithidia luciliae* assay. Black bars represent A.TL (n = 16), and gray bars represent A.SW (n = 10) mice treated with AuTM. No controls are shown because the percentages were 0. Percentage of positivity was evaluated using Fisher’s exact test. (**B**) Anti-chromatin, -histones, and -RNP antibodies in A.TL and A.SW mice during 15 weeks of AuTM or H_2_O (control) treatment detected by ELISA. Circles represent mean values of A.TL mice treated with AuTM (black, n = 16) or control (empty circle, n = 14). Squares represent mean values of A.SW mice treated with AuTM (gray, n = 10) or control (empty square, n = 9). Error bars indicate SEM. * = *p* < 0.05; ** = *p* < 0.01; *** = *p* < 0.001, **** = *p* < 0.0001 indicate differences between A.TL and A.SW mice treated with AuTM. & = *p* < 0.05; && = *p* < 0.01; &&& = *p* < 0.001, &&&& = *p* < 0.0001 indicate differences between A.TL mice treated with AuTM and controls using the Mann–Whitney U test.

**Figure 4 biology-13-00812-f004:**
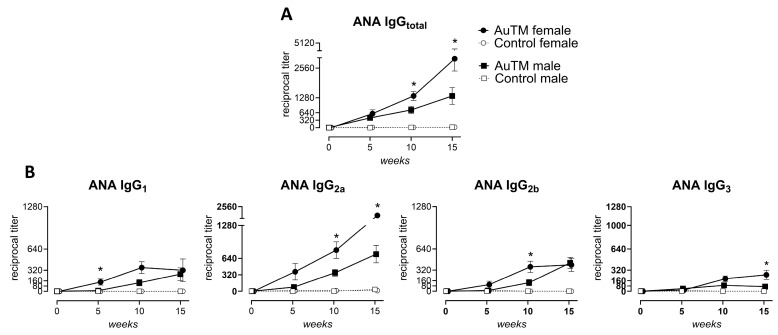
Differences in ANA titer between A.TL males and females during 15 weeks of AuTM treatment. (**A**) Titer of serum ANA of IgG_total_ and (**B**) IgG subclasses in A.TL males and females detected by indirect immunofluorescence staining of HEp-2 cells. Circles represent mean values of A.TL females treated with AuTM (black, n = 8) or controls (empty circle, n = 7). Squares represent mean values of A.TL males treated with AuTM (black, n = 8) or controls (empty square, n = 7). * = *p* < 0.05 indicate differences between A.TL males and females treated with AuTM using the Mann–Whitney U test. Error bars indicate SEM.

**Figure 5 biology-13-00812-f005:**
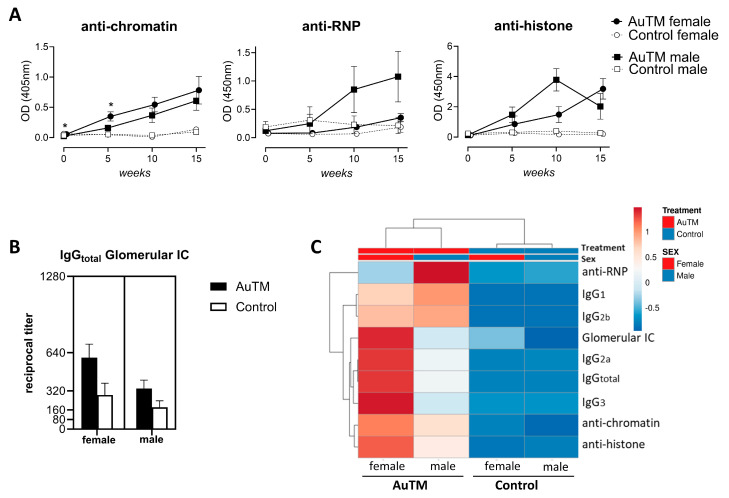
Differences in serum autoantibodies between A.TL males and females during 15 weeks of AuTM treatment. (**A**) Serum autoantibodies against chromatin, histones, and RNP antigens in A.TL males and females detected by ELISA. Circles represent mean values of A.TL females treated with AuTM (black, n = 8) or controls (empty circle, n = 7). Squares represent mean values of A.TL males treated with AuTM (black, n = 8) or controls (empty square, n = 7). * = *p* < 0.05 indicates differences between A.TL males and females treated with AuTM using the Mann–Whitney U test. Error bars indicate SEM. (**B**) IgG_total_ titer in glomerular immune complex deposition found in A.TL females and males after AuTM treatment and controls for 15 weeks. Black bars represent AuTM-treated mice (females = 8, males = 8), and empty bars represent control mice (females = 7, male = 7). (**C**) Heatmap of antibodies immune response at week 15 in A.TL mice. Columns with similar annotations (treatment and sex) were collapsed by taking a mean inside each group. Rows are centered; unit variance scaling is applied to rows. Both rows and columns are clustered using Euclidean distance and average linkage. A heatmap was performed using Clustvis software. Heatmap data matrix visualizes the values in the cells using a color gradient, which gives an overview of the largest and smallest values in the matrix.

**Table 1 biology-13-00812-t001:** H-2 haplotypes and number of AuTM-treated and control mice from each strain and gender.

Strain	H-2 Haplotype	H-2 Loci								Gender	AuTM	Control
		K	A_b_	A_a_	E_β_	E_a_	S	D	L		n	n
A.TL	t1	s	k	k	k	k	k	d	d	Male	8	7
										Female	8	7
A.SW	s	s	s	s	(s	s) ^a^	s	s	s	Male	5	5
										Female	5	5

^a^ No expression of E molecules on the cell surface [35].

**Table 2 biology-13-00812-t002:** Presence and titer of anti-nuclear (ANA) and anti-nucleolar (ANoA) antibodies and renal immune complex deposits, and presence of ANA of IgG subclasses and ds-DNA antibodies in A.TL and A.SW mice.

Strain	Treatment	Treatment Time (Weeks)	Fraction of IgG_total_ (Pos./Total No)	IgG Pattern (No)	* ANA Titer IgG_total_	Fraction of Subclasses (Pos./Total No)				Fraction of dsDNA (Pos./Total No)	* IgG_total_ Titer Glomeruli
						IgG_1_	IgG_2a_	IgG_2b_	IgG_3_		
A.TL	H_2_O (n = 14)	0	0	-	0	ND	ND	ND	ND	0	ND
		5	1/14	FSp (1)	80 ± 0	0	1/14	0	0	0	ND
		10	2/14	FSp (2)	80 ± 0	1/14	2/14	0	0	0	ND
		15	4/14	FSp (4)	80 ± 0	0	4/14	0	0	0	234 ± 56
	AuTM (n = 16)	0	0	-	0	ND	ND	ND	ND	0/16	ND
		5	15/16 ^d^	H (9), FSp (6)	544 ± 88 ^d^	6/16 ^a^	8/16 ^b^	4/16	6/16 ^a^	4/16	ND
		10	16/16 ^d^	H (13), FSp (3)	1060 ± 142 ^d^	12/16 ^d^	15/16 ^d^	13/16 ^d^	11/16 ^d^	11/16 ^d^	ND
		15	16/16 ^d^	H (12), FSp (4)	2160 ± 372 ^d^	8/16 ^b^	16/16 ^d^	14/16 ^d^	10/16 ^c^	11/16 ^d^	470 ±72 ^b^
A.SW	H_2_O (n = 9)	0	0	-	0	ND	ND	ND	ND	0	ND
		5	0	-	0	ND	ND	ND	ND	0	ND
		10	0	-	0	ND	ND	ND	ND	0	ND
		15	0	-	0	ND	ND	ND	ND	0	80 ± 37
	AuTM (n = 10)	0	0	-	0	ND	ND	ND	ND	0	ND
		5	6/10 ^a^	cp (6)	200 ± 40 ^c^	3/10	5/10 ^a^	2/10	0	0	ND
		10	10/10 ^d^	cp (10)	480 ± 160 ^d^	7/10 ^b^	10/10 ^d^	8/10 ^c^	1/10	0	ND
		15	10/10 ^d^	cp (10)	832 ± 128 ^d^	9/10 ^c^	10/10 ^d^	8/10 ^c^	8/10 ^c^	1/10	160 ± 86

n: number of mice, cp: clumpy; H: homogenous; FSp: fine speckled pattern, ND (no data), * Mean of IgG_total_ titer of positive individuals ± SEM. ^a,b,c,d^ = *p* value ≤ 0.05, 0.01, 0.001, 0.0001 indicates differences to week 0 using the Mann–Whitney U test. Positivity of individuals compared to week 0 was evaluated using Fisher’s exact test.

**Table 3 biology-13-00812-t003:** The total number of splenocytes and the B and T cell markers in AuTM-treated and control A.TL and A.SW female mice at week 15, analyzed by flow cytometry.

Strain	Treatment	No. of Splenocytes	CD19^+^	CD3^+^	CD4^+^, CD25^+^	CD3^+^, CD90.2^+^	CD3^+^, CD49b^+^, c-IFN-γ^+^
A.TL	H_2_O (n = 4)	12.08 ± 2.18	5.18 ± 0.92	2.55 ± 0.44	0.11 ± 0.02	2.05 ± 0.34	0.01 ± 0.00
	AuTM (n = 5)	14.52 ± 2.10	9.75 ± 0.52 ^a^	2.9 ± 0.42	0.10 ± 0.2	1.99 ± 0.18	0.03 ± 0.02
A.SW	H_2_O (n = 5)	10.64 ± 0.47	4.47 ± 0.26	4.92 ± 0.12	0.97 ± 0.05	0.74 ± 0.12	0.15 ± 0.1
	AuTM (n = 5)	17.52 ± 0.74 ^b^	8.14 ± 0.48 ^b^	7.24 ± 0.83 ^a^	0.51 ± 0.12	1.42 ± 0.19 ^a^	0.56 ± 0.4 ^b^

n: number of mice. Figures represent mean number of cells (×10^6^) ± SEM, ^a,b^ = *p* value ≤ 0.05, 0.01; compared to control (H_2_O) using the Mann–Whitney U test.

**Table 4 biology-13-00812-t004:** Presence and titer of anti-nuclear antibodies (ANA), renal immune complex deposits, and presence of ANA of IgG subclasses and ds-DNA antibodies in A.TL females and males treated with AuTM.

A.TL Gender	Treatment Time (Weeks)	Fraction of IgG_total_ (Pos./Total No)	IgG Pattern (No)	* ANA Titer IgG_total_	Fraction of Subclasses (Pos./Total No)				Fraction of dsDNA (Pos./Total No)	* IgG_total_ Titer Glomeruli
					IgG_1_	IgG_2a_	IgG_2b_	IgG_3_		
Male (n = 8)	0	0/8	-	0	ND	ND	ND	ND	0/8	ND
	5	8/8 ^c^	H (6), FSp (2)	420 ± 67 ^c^	1/8	3/8	1/8	4/8	1/8	ND
	10	8/8 ^c^	H (6), FSp (2)	760 ± 160 ^c^	5/8 ^a^	8/8 ^c^	5/8 ^a^	4/8	5/8 ^a^	ND
	15	8/8 ^c^	H (7), FSp (1)	1360 ± 366 ^c^	5/8 ^a^	8/8 ^c^	8/8 ^c^	3/8	5/8 ^a^	340 ± 70
Female (n = 8)	0	0/8	-	0	ND	ND	ND	ND	0/8	ND
	5	7/8 ^b^	H (3), FSp (4)	685 ± 162 ^c^	5/8 ^a^	5/8 ^a^	3/8	2/8	3/8	ND
	10	8/8 ^c^	H (7), FSp (1)	1360 ± 188 ^c^	7/8 ^b^	7/8 ^b^	8/8 ^c^	7/8 ^b^	6/8 ^b^	ND
	15	8/8 ^c^	H (5), FSp (3)	2960 ± 526 ^c^	3/8	8/8 ^c^	6/8 ^b^	7/8 ^b^	6/8 ^b^	600 ± 112

n: number of mice, H: homogenous; FSp: fine speckled pattern, ND (no data). * Mean of IgG_total_ titer of positive individuals ± SEM. ^a,b,c^ = *p* value ≤ 0.05, 0.01, 0.001; indicates differences to week 0 using the Mann–Whitney U test. Positivity of individuals compared to week 0 was evaluated using Fisher’s exact test.

## Data Availability

Data supporting the findings of this study are available within the article or available from the corresponding author (S.H.) upon reasonable request.

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
