# Peer review of "Exposure to Gold Induces Autoantibodies against Nuclear Antigens in A.TL Mice"

_biology, 2024, doi:10.3390/biology13100812_

Round 1

Reviewer 1 Report

Comments and Suggestions for Authors

This is the review of the article entitled “Exposure to gold induces autoantibodies against nucleosome antigens in A.TL mice” by Sara Puente-Marin and Said Havarinasab.

Summary of the present work:

The authors exposed two mouse strains congenic for non-H-2 genes calles A.TL and A.SW to gold aurothiomalate for a time period going from 5 weeks to 15 weeks. Using different techniques, they evaluated different features of autoimmune diseases including anti-nuclear antibodies (DNA, Chromatin, histones and RNP). They also evaluated by flow cytometry the impact on immune cell distribution in the spleen as well as the sex influence on the immune responses. The authors demonstrated a stronger response in A.TL mice compared to A.SW especially in females.

General comment:

Despite the topic and the results are interesting, the manuscript could be improved to facilitate the understanding of global message of this work (especially the figures/tables that should be presented differently). The material and method part should be also more developed especially on controls, the dilutions used, the replicates etc… as well as the legends. Control mice used is also quite questioning. Results should be edited and organized to include some missing information and to remove the redundant one.

Specific comments:

Materials and Methods section:

Animal Cohort:

-There is a problem with the number of A.SW mice as it is mentioned in the text (L84) 19 animals with 10 males and 10 females.

-The A.TL mice are aged from 12 to 16 weeks while the A.SW mice are younger (8-12 weeks). Why not choosing a cohort of animals with the same age range? Can we fully exclude any influence of this age on the observed results? Have the authors considered to investigate if this parameter may be also involved (correlation?)?

-As it is not clearly indicated in the manuscript, I was wondering if the authors perform several independent experiments (with A.TL and A.SW being treated together) or is it an “A.TL” experiment compared to “S.SW” experiment or it is one experiment with all the animals.

-The control mice are questionable as they did not receive any injection of diluent? Both groups of animals should receive the same protocol.

Gold Treatment:

-Mice are receiving 0.45mg AuTM. Have the authors tried other quantity? As the authors are referring to other metals such as Mercury and Silver in their discussion, have they considered to use those metals in the set of experiments with AuTM to really validate the comparison they are making?  

-Regarding the ANA test, 

Can the authors define what is a specific nuclear staining? control antibodies were run? Could the authors explain what is a “young ANA negative mice”? why not using regular mice? Could the authors define if tests were performed in triplicate, duplicate? How the dilution 1:80 was defined? Is it not too high to be considered as “negative”?

-Regarding the anti-dsDNA test,

How was determined the dilution 1:10? Why no negative control was used? How many replicates were performed for each test?

-Regarding the anti-chromatin ELISA,

How was determined the dilution 1:400? How many replicates were performed for each test? Could the authors explain what is a “young ANA negative mice”? why not using regular mice?

-Regarding the anti-RNP ELISA

How was determined the dilution 1:200? How many replicates were performed for each test?

-Regarding the anti-histone ELISA

Was the background subtracted? How was the announced dilution 1:80 being optimized? Why control mice were not used? How many replicates were performed for each test?

=> General comment, for all assays including ELISA/ELISA-derived techniques:

Were all samples (from the whole study) analyzed at once? a same plate? As it seems that there is no standard in those assays, how did the authors manage “experimental batch effect”? Even though, the approach is well-standardized in terms of procedure, it is very hard to guarantee, especially considering the OD was used as the readout, that there is no variation between experiments, plates etc…

-Regarding flow cytometry approach,

Why the authors are using rabbit antibodies as FC blocking reagent? they should have used mouse antibody blocking reagent.

Why CD45 is not used to identify leukocytes?

The authors are using redundant fluorochrome on various antibodies which is perfectly fine when you have exclusive markers (ie CD3/CD4/CD25/CD19) but when it comes to PE-CD49b and PE-CD90.2, CD3+ cells can be express on both cell populations so how can we distinguish between CD3+CD49b+ and CD3+CD90.2+?

A presentation of the gating strategy (and the dublet removal) and an example of an obtained profile is mandatory to facilitate the understanding on how each population is define.

The IFNg staining procedure described in the results does not appear anywhere? Is it intracellular staining? Secretion assay? ELISA? Is it spontaneous or induced secretion ?

-Regarding the glomerular immunocomplex deposits,

How was the threshold set to 1:40? How many replicates were performed for each animal?

Results Section:

Figure 1: Title and Legend should be edited to be more precise. Images obtained with control mice receiving the same experimental procedure should appear.

Table 2: it should appear before Figure 2. The table is carrying a lot of information, some of them are represented as graphs already. Some information are redundant as the “Fraction of IgG…” vs. “Total IgG (%)”?

How do the authors explain the appearance of FSp positive mice during the time course and glomeruli IgG in untreated A.TL mice? Is it a sign of age/spontaneous induction of autoimmune diseases? Could this be explained the age difference of A.SW ones? Have they tried to do a longer time course with those A.SW mice (up to 20-25 weeks)?

Supplementary Figure 1: Title could be improved to be more precise. Bigger title for each mouse strain should be used “A.TL” and “A.SW”.

In the figure, the dilution 1:80 considered by the authors as “negative” should appear in the graph and not the value “0” as they did not go further in the dilution serie.

Figure 2: Title and legend description should be improved. Supplementary Figure 1 should be merged with it. Authors should use asterix to indicate statistical significance.

Figure 3: Title and legend description should be more precise. Results obtained with control mice should appear on graphs. Authors should use asterix to indicate statistical significance.

Figure 4: The figure is redundant with the third one, both figures should be merged (including anti-dsDNA with controls)

Table 3: Title and legend should be improved. Why did the authors decide to investigate the immune cell distribution of female mice only? This aspect should be clarified. Instead of a table, those results could be represented by a graph. This part could be placed at the end of this work? CD4+CD25+ should not be considered as T reg only, they can be activated T cells. Have the authors investigated B cell activation through the expression of CD25? Have other cytokines such IL-12, IL-6, IL-17, TNFa been measured in cells, serum etc?

Figure 5: Title and figure legend should be improved. Figure size should be increased. Distribution of results/titers should be compiled as in Figure 4. Results obtained with control mice should appear. The section 5D with the heatmap brings the results from control mice but does not bring much information than the ones found on the graphs (or poorly explored by the authors)

Table 4: It is carrying redundant information and should be simplified. Some results are already shown in Figure 5. Results obtained with control mice should appear.

Discussion Section:

L365: The authors are mentioning that Mercury is stimulating T cells through the production of cytokines? Is it a direct effect? Do the authors have evaluated the presence of various cytokines (apart the IFNγ briefly mentioned)? Have they considered to evaluate the impact of AuTM on T and B cell activation, proliferative response and cytokine production in simple in vitro models.

L388: B220 should not be used as it evolves depending on B cell differentiation (decrease in plasma cells) and can be present on dendritic cells, T cells…

L391: The statement “However, no increase in the number of T cells… B cells are not stimulated through T cells… A.SW mice” should be modified as a stimulation through T cells does depend on their number only but also their state of activation (expression of costimulatory molecules, production of cytokines etc).

L394: As the authors are comparing their results to those obtained with Mercury, have they considered to use this metal as “positive control” for their study. Maybe the observed differences (lower titers in antibodies) come from a “experimental batch effect” and this could lead to a questioning about the robustness of the model?

Further in their discussion, the authors mentioned the use of SW mouse but why they did not create a group of mice in those experiments? but have they considered to use other strains of mice? especially ones that do not develop auto immune diseases? Also, have they tried to expose MRL/I, NZB/NZW mice to AuTM to see if it induces the features of the auto immune diseases more rapidly.

Author Response

Dear Reviewer 1, 

Thank you for taking the time to read and review our manuscript. Please find attached the document with the response to your comments. You can also find the new version of the manuscript with track changes in the re-submitted files. 

Reviewer 2 Report

Comments and Suggestions for Authors

Dear Author,

This research manuscript presented that mice strain congenic for non-H-2 genes, A.TL ((H-2tl) and A.SW (H-2s) ) can be induced for autoimmune disease using Gold aurothiomalate. Further, the author concluded that A.TL-exposed mice would be a model reference for autoimmune studies. 

Although the results are fascinating, they go counter to traditional Chinese and Indian understanding of the use of gold to cure or lessen the symptoms of autoimmune disorders. I think the data should be reinterpreted in the context of how gold affects immunity, and the researchers should offer additional proof for their theory that gold induces autoimmune diseases.

It would also be beneficiary for the paper to indicate how long the induced mice would be beneficial for autoimmune research, what sort of study might be conducted, and how long the mice would need to be induced for autoimmune disease.

regards and best wishes

Author Response

Dear Reviewer 2, 

Thank you very much for taking the time to read and review our manuscript. Please find attached the document with the responses to your comments. 
